# Effect of Batchelor Flow on Polymorphic Crystallization in a Rotating Disk Crystallizer

**Zun-Hua Li, Jinsoo Kim and Woo-Sik Kim \***

Functional Crystallization Center, Department of Chemical Engineering (Integrated Engineering Program), Kyung Hee University, Seocheon-Dong 1, Giheung-gu, Yongin-si 446-701, Korea; zhli@khu.ac.kr (Z.-H.L.); jkim21@khu.ac.kr (J.K.)

\* Correspondence: wskim@khu.ac.kr; Tel.: +82-31-201-2970; Fax: +82-31-273-2971

**Abstract:** In this work, the influence of Batchelor flow on the polymorphic crystallization in a rotating disk (RD) crystallizer was investigated. By regulating crystallization parameters, i.e., the rotation speed, cooling rate, and ethanol fraction, we found that a higher fraction of L-histidine stable Form-A at the induction time and a faster rate of phase transformation could be obtained in the RD crystallizer as compared to previous results in a mixing tank crystallizer. Based on these results, we concluded that the polymorphic crystallization in the RD crystallizer was more effective due to Batchelor flow fluid motion.

**Keywords:** Batchelor flow; rotating disk crystallizer; selective nucleation; phase transformation





## 1. Introduction

Currently, polymorphism is of great interest to the pharmaceutical industry due to the distinct properties of polymorphs, including their bioavailability, solubility, and stability [1–3]. Therefore, polymorphic crystallization plays a critical role in harvesting desired polymorphs for pharmaceutical applications. Previous studies on polymorphism in the crystallization process have demonstrated that experimental conditions, such as the rotation speed, supersaturation, solvent, cooling rate, ultrasound, and additives (seeding), are influential parameters affecting the resultant polymorphs in the process of polymorphic crystallization [4–9]. Hydrodynamics frequently plays a key role in these parameters during polymorphic crystallization. The stable form of carbamazepine crystals nucleated under the agitated state. However, metastable-form crystals were merely generated under the quiescent state [10]. Stirring can not only improve the nucleation of the stable phase of L-glutamic acid but can also enhance the phase transformation rate [11,12]. Furthermore, it was also demonstrated that the hydrodynamics pattern and stirring speed can affect the polymorphic nucleation [13].

We have previously reported the efficacy of the Batchelor flow in co-crystallization [14] and spherical agglomeration [15] and that of the Taylor flow in the reaction [16–18], antisolvent [19,20], and cooling crystallization [21–25], which, in turn, concern the stoichiometrically diverse co-crystals, crystal agglomeration, polymorphic nucleation, phase transformations, crystal size distributions, and crystal shapes. Hence, this work intends to study the effect of periodic Batchelor flow on the polymorphic crystallization in a rotating disk (RD) crystallizer. The Batchelor flow fluid motion was generated in fluid [26], which consisted of a Bödewadt layer and a von Kármán layer [27,28]; the Batchelor flow can be classified into four types of flow regimes [29–31]. In this study, the RD crystallizer exhibited laminar or turbulent Batchelor flow regimes, which were highly dependent on the rotation speed. This type of reactor possesses good scalability for industrial applications, such as its high mass transfer. For example, Visscher et al. demonstrated that the mass transfer rate of liquid–liquid in an RD reactor was ten times higher than that in a packed column and art microchannel [32]. Moreover, the liquid–solid mass transfer rate in the RD reactor

was much higher than that in a packed bed reactor [33]. According to the above literature reports, it is logical to speculate that the application of a stator–rotor cavity reactor will also greatly improve the crystallization efficiency, such as the phase transformation and nucleation.

Herein, we employed an RD crystallizer instead of a traditional crystallizer to investigate the effect of Batchelor flow on the polymorphic crystallization at varying rotation speeds. In addition, the role of crystallization parameters, such as the cooling rate and ethanol fraction, was also investigated to comprehensively analyze the effect of the RD crystallizer performance.

## 2. Materials and Methods

As shown in Figure 1, the RD crystallizer was equipped with thermal jackets on the rotor and stator. In addition, a heating and cooling circulator was used for temperature control, and a thermocouple was used for temperature monitoring. The reaction space of the RD crystallizer between the rotor and stator was around 420 cm$^3$. L-histidine solid (Sigma-Aldrich, Saint Louis, MO, USA, stable Form-A, purity $\geq$ 99%) was dissolved in a solvent at a temperature that was 5 °C higher than the saturation temperature to prepare the feed solution. The ethanol fraction was adjusted from 0 to 30, 40, and 50 $v/v$% for the four corresponding samples. In addition, the initial concentration of the feed solution was set to 7.00, 2.00, 1.50, and 0.75 g/100 g solvent, respectively. Ethanol was obtained from Samchun Chem. (purity $\geq$ 99.5%, Seoul, Korea), and the water was membrane-purified (Applied Membranes Inc., Vista, CA, USA). The prepared feed solution was then pumped into the preheated RD crystallizer by the peristaltic pump. The rotor was rotated by the motor for half an hour to stabilize the system and to generate Batchelor flow fluid motion. The flow pattern in the RD crystallizer followed laminar Batchelor flow or turbulent Batchelor flow depending on the rotation speed. However, both flow patterns exhibited periodic fluid motion. Furthermore, the rotation speed was changed from 300 to 500, 700, 1000, and 1500 rpm. Moreover, the cooling rate was varied from 10 to 30, 150, and 360 °C/h. The initial and final setting temperatures were 50 and 10 °C, respectively. After reaching an extremely fast cooling rate of 360 °C/h, the feed solution in the RD crystallizer was first heated to the initial setting temperature of 50 °C by one heating and cooling circulator, after which the RD crystallizer was quickly shifted to another heating and cooling circulator, which cooled the solution to 10 °C. When the feed solution reached 10 °C, the cooling rate was calculated by evaluating the cooling time. Crystallization was continued for 500 min after the temperature in the RD crystallizer reached the final setting temperature.

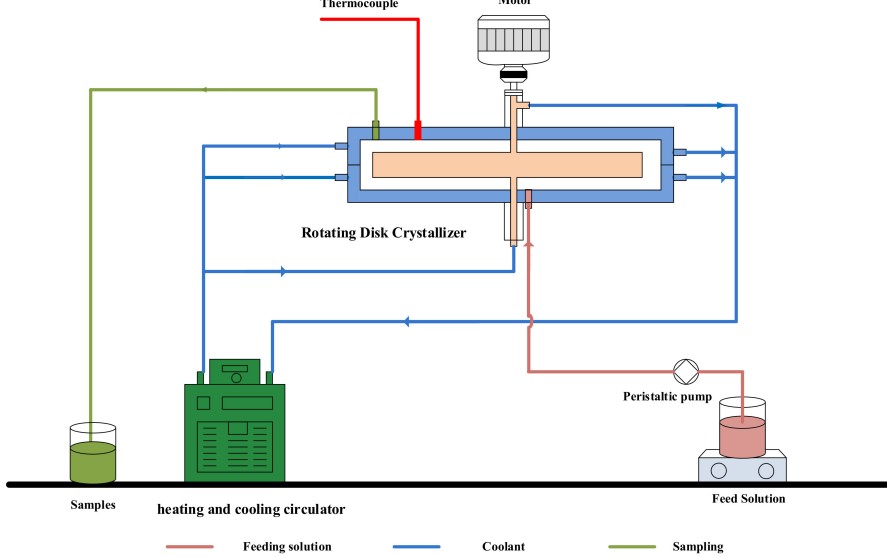

**Figure 1.** Schematic diagram of a rotating disk (RD) crystallizer for cooling crystallization.

In order to monitor the entire crystallization process, the crystal samples suspended in the solution were intermittently collected. The suspension was filtered, and then the obtained powder was dried for 12 h in an oven at 60 °C. In the process of sampling, the prepared feed solution was simultaneously refilled into the RD crystallizer to maintain the experimental consistency of the Batchelor flow fluid motion in the crystallization process. All collected crystal samples were characterized by powder X-ray diffraction (PXRD; Bruker, D8 Advance, Karlsruhe, Germany) and Fourier-transform–Raman-spectroscopy (FT-Raman; Renishaw, Gloucestershire, UK). The polymorph calibration of the L-histidine crystals followed a previously established protocol [25].

## 3. Result and Discussion

### 3.1. Effect of Rotation Speed

The effect of rotation speed on polymorphic crystallization was studied by changing the rotation speed from 300 to 1500 rpm in the RD crystallizer, and the results are shown in Figure 2. In order to ensure the repeatability of the experiment, each experiment was performed in triplicate. The induction time ($t_I$) at different rotation speeds is indicated by the different colored vertical dotted lines, and the fraction of L-histidine stable Form-A at the induction time ($A_I\%$) at different rotation speeds is indicated by the different colored horizontal dotted lines. It was demonstrated that the $t_I$ decreased from 112 ( black vertical dotted line) to 52 min (pink vertical dotted line), while the $A_I\%$ increased from approximately 35% (black horizontal dotted line) to 76% (pink horizontal dotted line) when the rotation speed increased from 300 to 1500 rpm. After the induction, the crystallization process was followed by the phase transformation, and the fraction of L-histidine stable Form-A was increased at each rotation speed following the crystallization time. Finally, the fraction of L-histidine stable Form-A was increased from about 54% to 100% as the rotation speed increased from 300 to 1500 rpm when the crystallization time was 500 min. It should be noted that the fraction of L-histidine stable Form-A reached 100% only when the rotation speed was higher than 1000 rpm. This indicated that samples obtained from those two higher rotation speeds were the pure stable Form-A of L-histidine. Interestingly, at the low rotation speeds, including 300, 500, and 700 rpm, the fraction of L-histidine stable Form-A was still increasing slightly when the crystallization time approached 500 min; therefore, it could be speculated that the fraction of L-histidine stable Form-A at 300, 500, and 700 rpm can also reach 100% after an extremely long crystallization time.

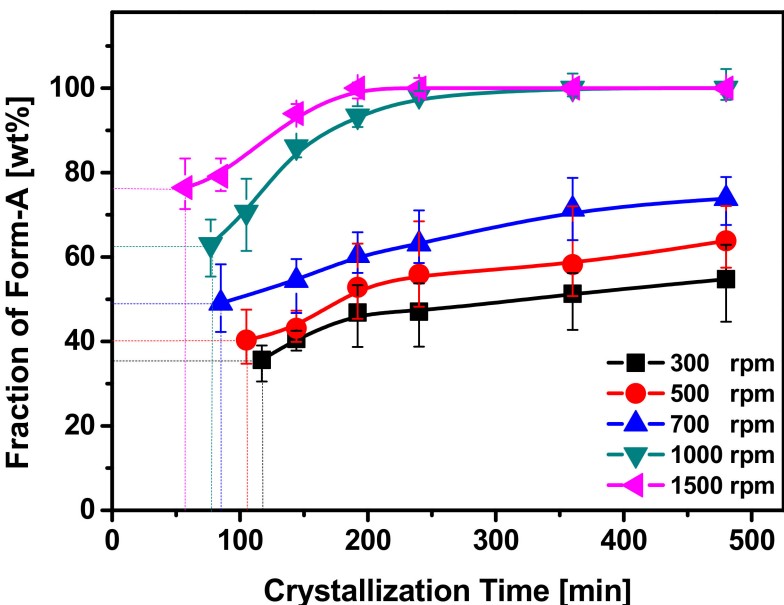

**Figure 2.** Fraction of the Form-A profile at different rotation speeds. The cooling rate was 10 °C/h, the ethanol fraction was 30%, and the induction time ($t_I$) is indicated by the dotted line.

Raman spectroscopy was performed on the L-histidine polymorphic samples to determine the L-histidine stable Form-A fraction. As shown in Figure 3a, the absorption peak that appeared at 215–245 cm$^{-1}$ can be defined as the characteristic peak of Form-A, and the polymorphic fraction of Form-A was proportional to the peak area ratio. Therefore, the fraction of Form-A could be obtained by the peak area ratio shown in Figure 3b, and by following the protocol previously reported by Park and Kim [25]. Accordingly, the weight fraction of the L-histidine stable Form-A quickly increased from 76% (57 min) to 100% (192 min) within 135 min at 1500 rpm. To further confirm the formation of the L-histidine stable Form-A, the PXRD results were used to analyze the samples obtained at different times, as shown in Figure 3c. According to Park and Kim [25], the diffraction peak at 17.75° was defined as the feature peak of L-histidine stable Form-A, and the diffraction peak at 17.28° was defined as the feature peak of L-histidine metastable Form-B. Therefore, as shown in Figure 3c, when the rotation speed was 1500 rpm, the feature peak intensity of L-histidine stable Form-A at 17.75° quickly increased as the crystallization time increased from 57 to 192 min. In comparison, the feature peak intensity of L-histidine metastable Form-B at 17.28° quickly decreased and finally disappeared. The results demonstrated that the sample obtained at 192 min was pure L-histidine stable Form-A.

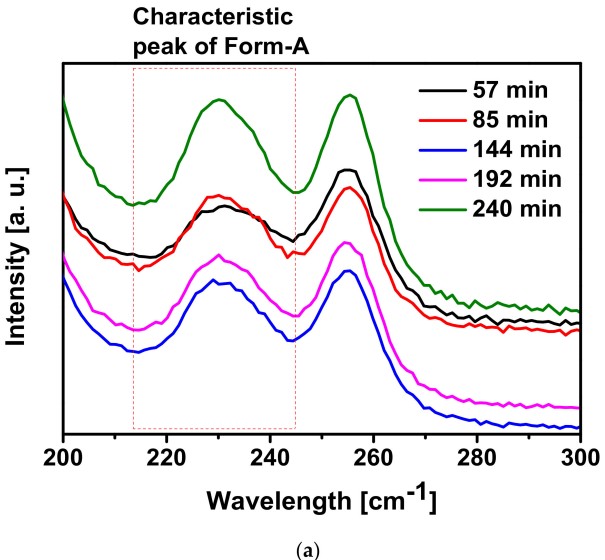

(a)

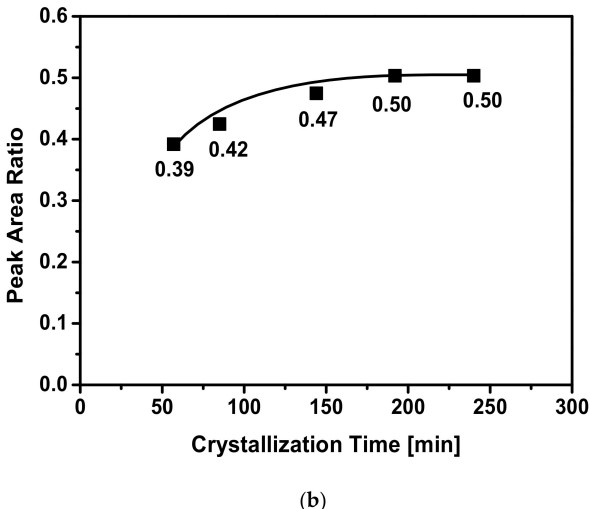

(b)

**Figure 3.** *Cont.*

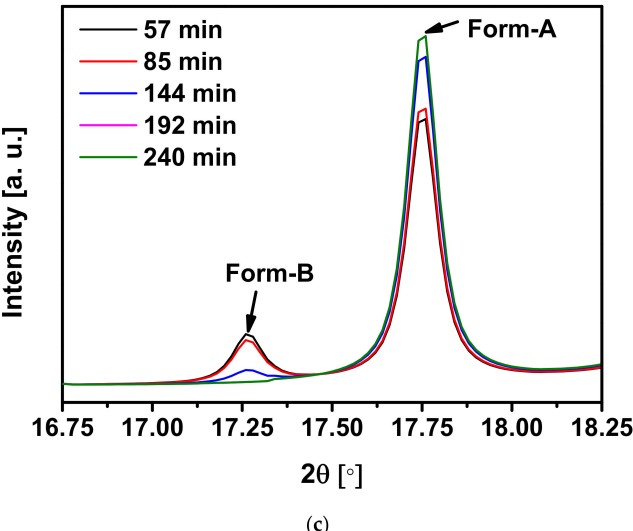

(c)

**Figure 3.** Characterization of the samples obtained during cooling crystallization at a rotation speed of 1500 rpm. (**a**) Raman spectroscopy, (**b**) peak area ratio of the samples obtained at different crystallization times, and (**c**) powder X-ray diffraction (PXRD) pattern. The cooling rate was 10 °C/h and the ethanol fraction was 30%. Peak area ratio = [(peak area of 215–245 cm$^{-1}$)]/[(peak area of 215–245 cm$^{-1}$) + (peak area of 245–270 cm$^{-1}$)].

As indicated in Figure 4a, the $A_I$% increased linearly from 35% to 76%, implying that higher rotation speeds were more efficient for the nucleation of L-histidine stable Form-A because a higher rotation speed provided a higher energy dissipation rate to enhance L-histidine stable Form-A nucleation. Moreover, the effect of rotation speed on the polymorphic nucleation was distinctly confirmed by analyzing the $t_I$ and induction temperature ($T_I$). Figure 4b shows the results following a linear $t_I$ decrease from 112 to 52 min and a $T_I$ increase from 31 to 41 °C. The higher rotation speed was more efficient during the shorter $t_I$ and at higher $T_I$. Therefore, it was implied that a higher rotation speed induced nucleation at a lower supersaturation level. Similarly, this lower supersaturation level may facilitate L-histidine stable Form-A nucleation, resulting in a high $A_I$%. In addition, as compared to the same experiments with an MT crystallizer, the $t_I$ was approximately 30 min earlier and the $T_I$ was approximately 5 °C higher in Batchelor flow [25]. As shown in Figure 2, 100% L-histidine stable Form-A was obtained at 192 min in the RD crystallizer at 1500 rpm, while a longer time (88.5 h) was required to achieve 100% L-histidine stable Form-A in the MT crystallizer [25].

As demonstrated in Figure 5, the rate of phase transformation was proportional to the rotation speed. The phase transformation rate was increased from 3.15 to 6.25 wt%/h. Normally, the phase transformation is directly related to the mass transfer. Therefore, the rate of phase transformation increased because of the acceleration of mass transfer caused by the increase in rotation speed. This result also demonstrated that mass transfer relies on rotation speed, which was confirmed by previous studies [20,22,25].

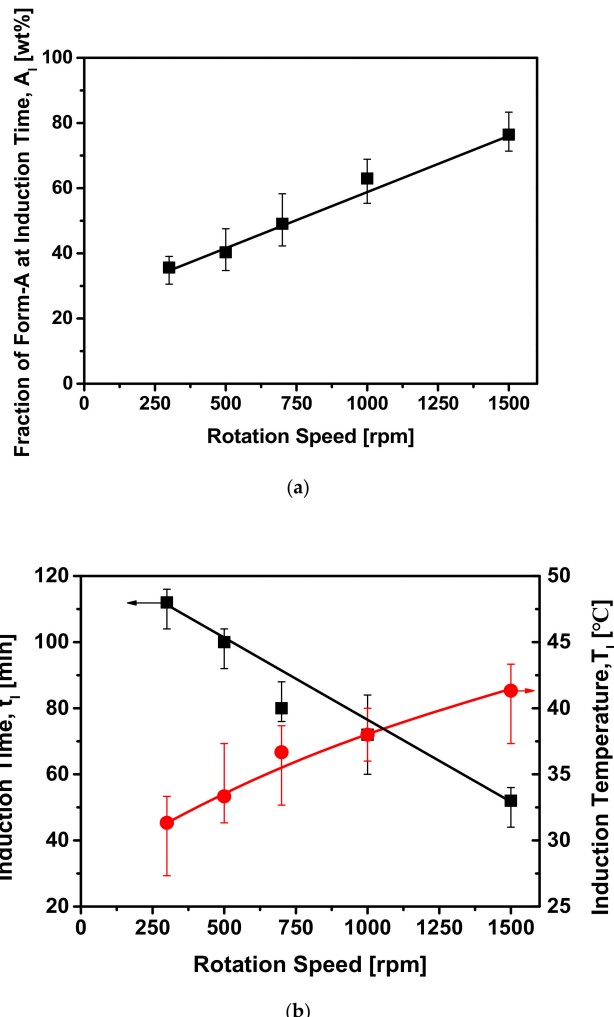

(a)

(b)

**Figure 4.** Effect of rotation speed on polymorphic nucleation: (**a**) $A_I$% and (**b**) $t_I$ and $T_I$. The cooling rate was 10 °C/h and the ethanol fraction was 30%.

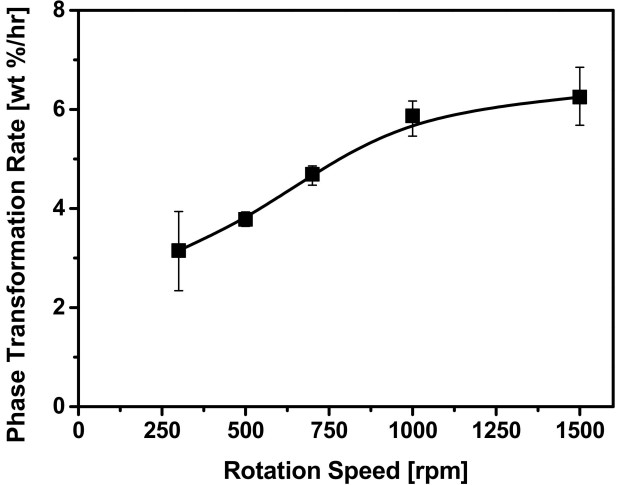

**Figure 5.** Effect of rotation speed on the rate of phase transformation. The cooling rate was 10 °C/h and the ethanol fraction was 30%.

### 3.2. Effect of Cooling Rate

The effect of cooling rate on the polymorphic crystallization of L-histidine was studied by changing the cooling rate from 10 to 360 °C/h. As shown in Figure 6a, the $A_I$% decreased from 40% to 20% as the cooling rate increased from 10 to 30 °C/h, which indicated that a slower cooling rate was more efficient for the nucleation of L-histidine stable Form-A. However, it should be noted that the $A_I$% was just shown a small change when the cooling rate was great than 30 °C/h, such as the 17% and 16% of $A_I$% was obtained at the cooling rate of 150 °C/h and 360 °C/h, respectively. It implied that a higher cooling rate has a small effect on enhancing the nucleation of L-histidine stable Form-A. Generally, in the pharmaceutical industry, a very high cooling rate, such as 150 or 360 °C/h, is difficult to achieve. Therefore, a slow cooling rate is a good choice for cooling polymorphic crystallization of the stable form. Moreover, the effect of cooling rate on polymorphic nucleation can be confirmed by measuring the $t_I$ and $T_I$ of nucleation. As shown in Figure 6b, the $t_I$ dramatically decreased from 90 to 3.6 min and the $T_I$ linearly decreased from 40 to 30 °C as the cooling rate increased from 10 to 360 °C/h. These results indicate that both the $t_I$ and $T_I$ were reduced as the cooling rate increased because the supersaturation level increased with the cooling rate. Therefore, a lower $A_I$% was induced by the higher initial supersaturation level.

Moreover, as indicated in Figure 7, the rate of phase transformation was highly dependent on the cooling rate. The phase transformation rate dramatically decreased from 3.83 to 1.05 wt%/h as the cooling rate increased from 10 to 30 °C/h. Moreover, it should be noted that the phase transformation rate only had a small change when the cooling rate was greater than 30 °C/h; for example, phase transformation rates of 0.60 and 0.44 wt%/h were achieved at the cooling rates of 150 and 360 °C/h, respectively. It implied that a higher cooling rate showed a smaller effect on the phase transformation rate of L-histidine.

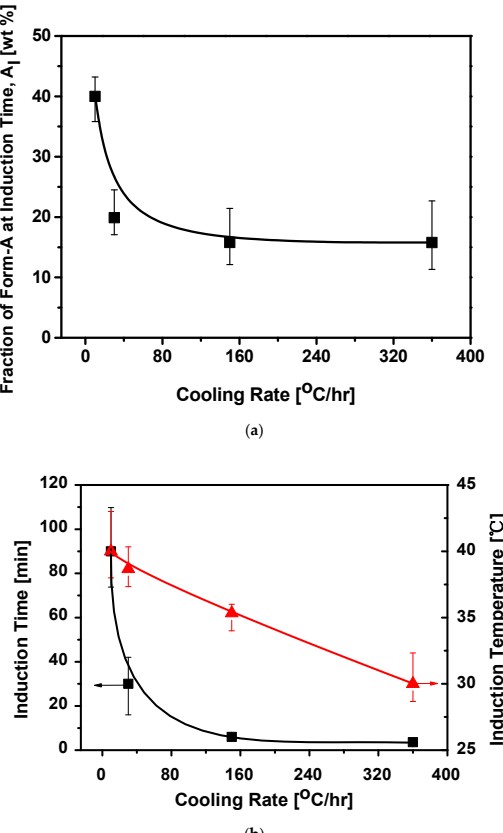

(a)

(b)

**Figure 6.** Effect of cooling rate on polymorphic nucleation: (**a**) $A_I$%, and (**b**) $t_I$ and $T_I$. The rotation speed was 500 rpm and the ethanol fraction was 30%.

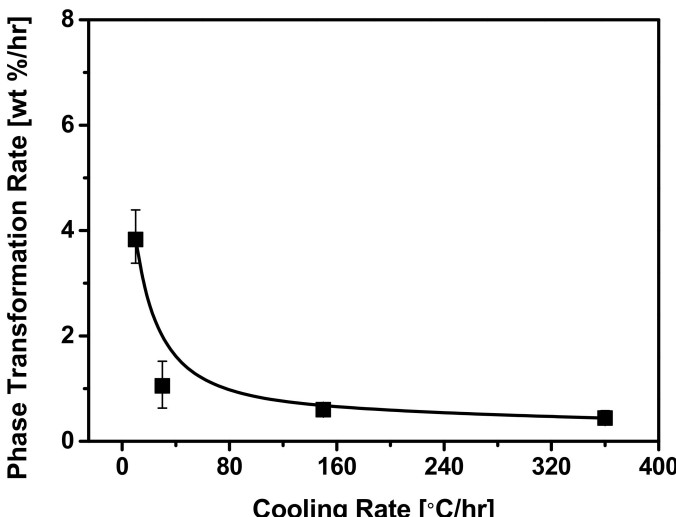

**Figure 7.** Effect of cooling rate on the phase transformation rate. The rotation speed was 500 rpm and the ethanol fraction was 30%.

### 3.3. Effect of Ethanol Fraction

The influence of ethanol fraction on polymorphic crystallization was studied by changing the fraction of ethanol from 0 to 50 vol% in the RD crystallizer, and the results are shown in Figure 8. The $t_I$ at different ethanol fractions is indicated by the different colored vertical dotted lines, and the $A_I\%$ at different ethanol fractions is indicated by the different colored horizontal dotted lines. It is indicated that the $t_I$ decreased from 108 (green vertical dotted line) to 48 min (black vertical dotted line), and the $A_I\%$ increased from 17% (green horizontal dotted line) to 71% (black horizontal dotted line) when the ethanol fraction was decreased from 50 to 0 vol%. With the increasing crystallization time, the fraction of L-histidine stable Form-A was increased at each ethanol fraction. Until the end of the experiment after 500 min, the fraction of L-histidine stable Form-A decreased from 100% to 28% as the ethanol fraction increased from 0 to 50 vol%. It should be noted that when the ethanol fraction was 0 vol%, the fraction of L-histidine stable Form-A was 100%, which showed that the sample was the pure stable Form-A of L-histidine at an ethanol fraction of 0 vol%. Interestingly, the fraction of L-histidine stable Form-A obtained at ethanol fractions of 30, 40, and 50 vol% still had a slight increase when the crystallization time approached 500 min. Therefore, we also can speculate that the fraction of L-histidine stable Form-A at 30, 40, and 50 vol% also can reach 100% with an extremely long crystallization time.

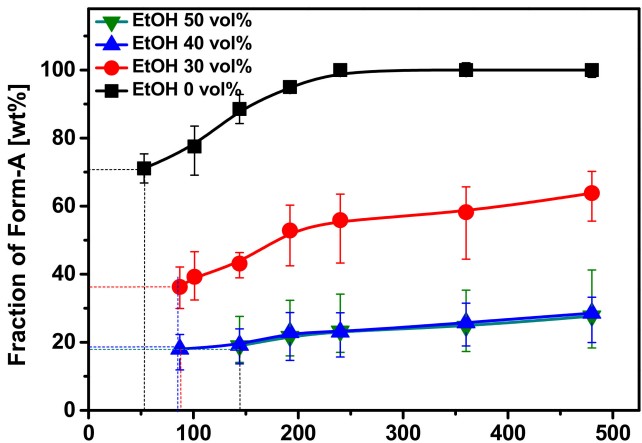

**Figure 8.** Fraction of the Form-A profile at different ethanol fractions. The rotation speed was 500 rpm and the cooling rate was 10 °C/h, and the induction time ($t_I$) is indicated by the dotted line.

The $A_I\%$ and the final fraction of L-histidine stable Form-A were dependent on the ethanol fraction. It can be seen from Figure 9a that the use of 0 vol% ethanol as the solvent produced an $A_I\%$ of approximately 71%, which then rapidly increased to 100% after 240 min at 500 rpm. The $A_I\%$ increased from 17% to 71% as the ethanol fraction was decreased from 50 to 0 vol%. Moreover, the $t_I$ increased from 48 to 108 min and the $T_I$ decreased from 42 to 32 °C as the ethanol fraction increased from 0 to 50 vol% (Figure 9b). In general, polymorphic crystallization is associated with the composition of the solvent because the solubility of the polymorphs is dependent on the mixed solvent. Therefore, the polymorph solubility of L-histidine was observed to be reduced as the ethanol fraction increased [25]. As a result, when the ethanol fraction increased, nucleation proceeded at a higher supersaturation level, thus producing a lower $A_I\%$. Moreover, the solubility difference is an actual driving force in the process of phase transformation, which dramatically increased when the ethanol fraction was decreased, leading to a fast phase transformation. Therefore, when the ethanol fraction was decreased, the solubility difference increased, resulting in a dramatic increase in the rate of phase transformation. As demonstrated in Figure 10, the rate of phase transformation increased from 1.47 to 8.13 wt%/h as the ethanol fraction decreased from 50 to 0 vol%.

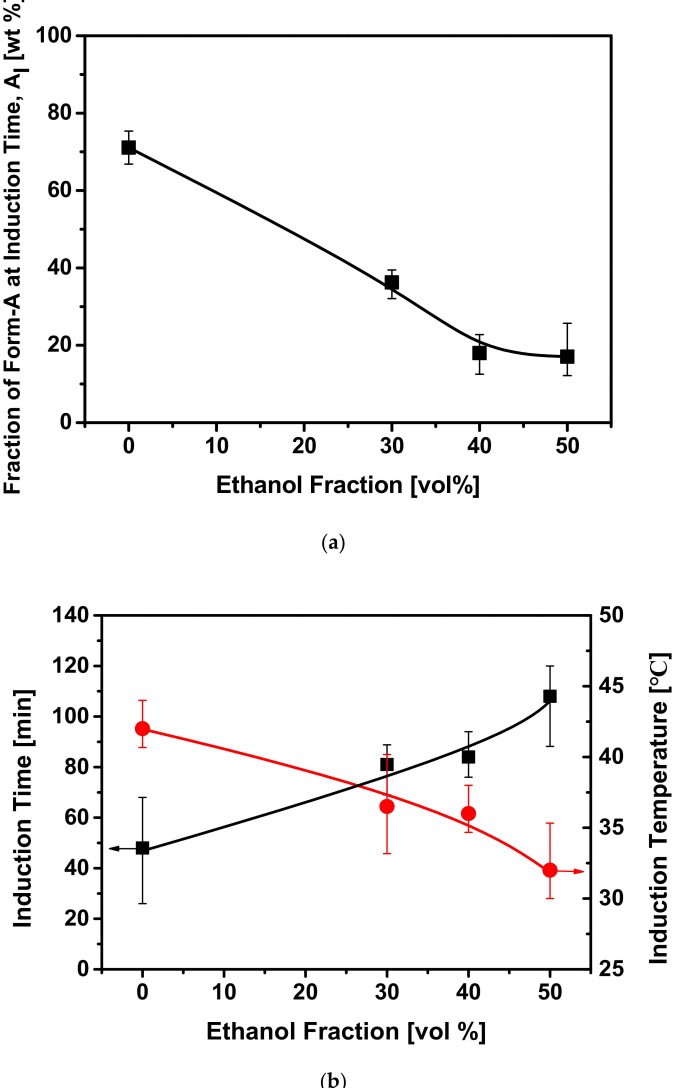

**Figure 9.** Effect of ethanol fraction on polymorphic nucleation: (**a**) $A_I\%$ and (**b**) $t_I$ and $T_I$. The rotation speed was 500 rpm and the cooling rate was 10 °C/h.

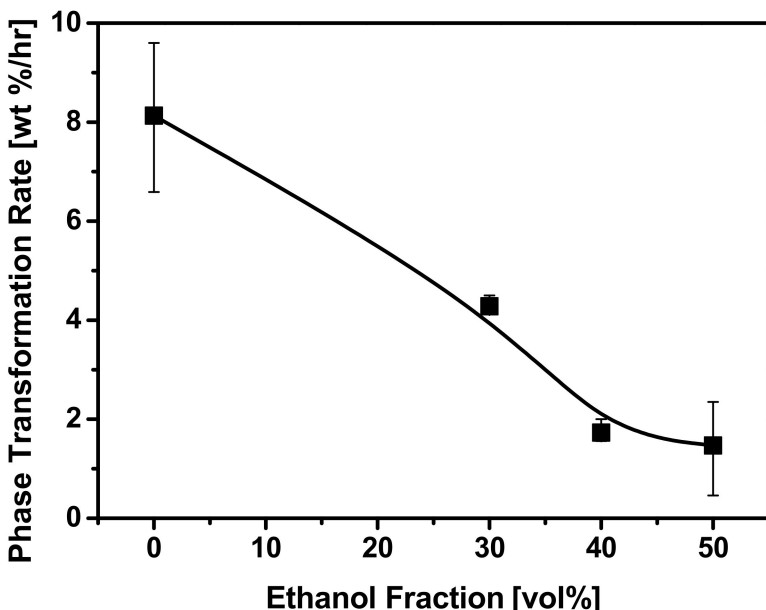

**Figure 10.** Effect of ethanol fraction on the phase transformation rate. The rotation speed was 500 rpm and the cooling rate was 10 °C/h.

## 4. Conclusions

The present study investigated the performance of an RD crystallizer in polymorphic crystallization of L-histidine by regulating the rotation speed, cooling rate, and ethanol fraction. The RD crystallizer dramatically facilitated the nucleation of stable Form-A due to the efficient Batchelor flow for mass transfer, and 100% Form-A was obtained after only 192 min at 1500 rpm. In addition, the cooling rate and ethanol fraction were also linked to polymorphic crystallization. The $A_I\%$ and the rate of phase transformation decreased as the cooling rate increased because the supersaturation level also increased. In addition, the induction supersaturation level was significantly dependent upon the ethanol fraction. Therefore, a lower $A_I\%$ was observed at a higher ethanol fraction. Moreover, an increase in the ethanol fraction dramatically decreased the rate of phase transformation due to the small solubility difference. These results show that polymorphic crystallization was more effective because of the role of Batchelor flow in the RD crystallizer.

**Author Contributions:** Conceptualization, Z.-H.L. and W.-S.K.; writing—original draft preparation, Z.-H.L.; writing—review and editing, J.K. and W.-S.K.; supervision, W.-S.K. All authors have read and agreed to the published version of the manuscript.

**Funding:** This research was funded by the Engineering Research Center of Excellence Program of the National Research Foundation of Korea (NRF) (Grant NRF-2021R1A5A6002853).

**Data Availability Statement:** Not Applicable.

**Acknowledgments:** Zun-Hua Li was financially supported by the China Scholarship Council.

**Conflicts of Interest:** The authors declare no conflict of interest.

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
