# Peer review of "Effect of Batchelor Flow on Polymorphic Crystallization in a Rotating Disk Crystallizer"

_crystals, doi:10.3390/cryst11060701_

Round 1

Reviewer 1 Report

This paper is so interesting for new technique of vortex effect in industrial crystallization. Especially, this paper describes effect of vortex on polymorph or phase transition of L-histidine crystals. Obviously, this study has shown that RD crystallizer is more effective than MT crystallizer. Therefore, this paper should be published in MDPI journal.

Author Response

Authors appreciate the kind comments of Reviewer-1, which must be helpful to improve our manuscript.

Comment: This paper is so interesting for new technique of vortex effect in industrial crystallization. Especially, this paper describes effect of vortex on polymorph or phase transition of L-histidine crystals. Obviously, this study has shown that RD crystallizer is more effective than MT crystallizer. Therefore, this paper should be published in MDPI journal.

Response: Thank you very much for the reviewers’ evaluation of our research.

Reviewer 2 Report

The manuscript ‘Effect of Batchelor Vortex Flow on Polymorphic Nucleation and Phase Transformation of L-histidine’ is studied the effect of Batchelor vortex flow mechanism using rotating disk and turbulent eddy flow mechanism using mixing tank crystallizers in producing the stable L-histidine polymorph.

Authors varied distinct parameters such as agitation or rotation speed, cooling rate and solvent ratios before drawing a conclusion of this study. It is known fact that the induction temperature and the % ethanol are inversely proportional to each other (due to distinct saturation levels), however it is interesting to see when it linked to the flow mechanisms. It concludes that the rate of phase transformation in the RD crystallizer is much higher than the MT crystallizer because of the effective mass transfer mechanism.

From this study, it can be seen that RD is more efficient than the MT type. Would this be limited to just this compound or can we expand it to the other systems as well? Saying Batchelor vortex flow is better to produce stable polymorphs?

When the temperature is constant at 10 °C (page 2 line 66), how authors studied the induction temperature about 40 °C (as per fig3. C)?

What device authors used to achieve this temperature (as the figures representing only chiller but not any other device to heat the solution)?

I could not see the starting solid form of L-Histidine (I mean the commercial material used for this study) in this manuscript.

I couldn’t find the samples sizes in the manuscript. Even though, Raman is enough for polymorph identification; am just wondering why authors do not consider testing these samples on PXRD just to make more robust? Is it due to the sample size?

I would like to see the FT-Raman spectra evidence (either in the supporting information or in the manuscript).

A suggestion is to move the sub-figure titles (a, b, c..) in to the middle of the figure (under the each diagram) to give more clarity.

Author Response

Authors appreciate the kind comments of Reviewer-2, which must be helpful to improve our manuscript.

The manuscript “Effect of Batchelor Vortex Flow on Polymorphic Nucleation and Phase Transformation of L-histidine” is studied the effect of Batchelor vortex flow mechanism using rotating disk and turbulent eddy flow mechanism using mixing tank crystallizers in producing the stable L-histidine polymorph.

Authors varied distinct parameters such as agitation or rotation speed, cooling rate and solvent ratios before drawing a conclusion of this study. It is known fact that the induction temperature and the % ethanol are inversely proportional to each other (due to distinct saturation levels), however it is interesting to see when it linked to the flow mechanisms. It concludes that the rate of phase transformation in the RD crystallizer is much higher than the MT crystallizer because of the effective mass transfer mechanism.

Comment 1: From this study, it can be seen that RD is more efficient than the MT type. Would this be limited to just this compound or can we expand it to the other systems as well? Saying Batchelor vortex flow is better to produce stable polymorphs?

Response 1: Yes, the Batchelor vortex flow is better to produce stable polymorphs, and it is can expand to other systems as well. For example, in another published paper in our lab that using the Batchelor vortex flow to study the cocrystallization of caffeine and maleic acid, the Batchelor vortex flow was favored to nucleate the stable 1:1 cocrystals than the metastable 2:1 cocrystals.

Comment 2: When the temperature is constant at 10 °C (page 2 line 66), how authors studied the induction temperature about 40 °C (as per fig3. C)?

Response 2: Sorry, that because we didn't express it clearly. Actually, the process of cooling crystallization is conducted by cooling down from 50 ℃ to 10 ℃ with different cooling rates. It was stated it clearly in the revised version.

Comment 3: What device authors used to achieve this temperature (as the figures representing only chiller but not any other device to heat the solution)?

Response 3: Sorry, that because we didn't express it clearly. Actually, during the experiment, we used a heating and cooling circulator to control the temperature. It was stated it clearly in the revised version.

Comment 4: I could not see the starting solid form of L-Histidine (I mean the commercial material used for this study) in this manuscript.

Response 4: Sorry, that because we didn't express it clearly. Actually, during the experiment, the starting solid form of L-Histidine was the stable Form A. It was stated it clearly in the revised version.

Comment 5: I couldn’t find the samples sizes in the manuscript. Even though, Raman is enough for polymorph identification; I am just wondering why authors do not consider testing these samples on PXRD just to make more robust? Is it due to the sample size?

Response 5: Sorry, we didn’t measure the sample particle size in this study. However, the PXRD testing can give a more robust result, so in the revised version, one example was given in the manuscript for illustration and validation with the PXRD pattern.

Comment 6: I would like to see the FT-Raman spectra evidence (either in the supporting information or in the manuscript).

Response 6: Yes, the Raman spectra evidence can give more robust result, so in the revised version, one example was given in this manuscript for illustration and validation with the Raman spectra.

Comment 7: A suggestion is to move the sub-figure titles (a, b, c..) in to the middle of the figure (under each diagram) to give more clarity.

Response 7: Thank you for your suggestion, the manuscript was revised according to your suggestion.

Reviewer 3 Report

This study investigated the polymorphic nucleation and transformation of L-histidine. Two reactors were used in this study and the RD crystallizer show superior characteristics in polymorph control. I have a few minor suggestions to this manuscript.

  1. In Introduction, please give comments on the scale up issue and future perspective for applying RD crystallizer in industry.
  2. In Experimental section, please specify the initial concentration of L-histidine and initial temperature in crystallization experiments at different ethanol/water concentration. In addition, please also give an illustration how to reach an extremely fast cooling rate of 360 oC/hr and why this extremely fast cooling rate was investigated. In my opinion, this extremely fast cooling rate is rarely used in industry. Furthermore, please also specify the volume of RD and MT crystallizer.
  3. In this study, only FTIR was used for solid-state property analysis, although calibration of polymorphs was presented in ref. [24], please at least give one example in this manuscript for illustration and validate with the PXRD pattern.
  4. In this study, in my opinion, adding ethanol in crystallization system seems no advantages for controlling the polymorph of L-histidine. Why this solvent was selected in this study?
  5. If possible, please also give examples to compare particle size and crystal habit of L-histidine from RD and MT crystallizer.
  6. Keywords are missing.

Author Response

Authors appreciate the kind comments of Reviewer-3, which must be helpful to improve our manuscript.

Comment 1: In Introduction, please give comments on the scale up issue and future perspective for applying RD crystallizer in industry.

Response 1: Thank you for your suggestion, some comments was given on the scale-up issue and future perspective for applying RD crystallizer in the industry according to your suggestion.

Comment 2: In Experimental section, please specify the initial concentration of L-histidine and initial temperature in crystallization experiments at different ethanol/water concentration. In addition, please also give an illustration how to reach an extremely fast cooling rate of 360 °C /hr and why this extremely fast cooling rate was investigated. In my opinion, this extremely fast cooling rate is rarely used in industry. Furthermore, please also specify the volume of RD and MT crystallizer.

Response 2: Thank you for your suggestion, in the revised manuscript, the initial concentration of L-histidine and initial temperature in crystallization experiments at different ethanol/water concentrations was specified. And also give an explanation of how to reach an extremely fast cooling rate of 360 ℃/h. Furthermore, the volume of RD and MT crystallizer in the revised version also specified.

Comment 3: In this study, only FTIR was used for solid-state property analysis, although calibration of polymorphs was presented in ref. [24], please at least give one example in this manuscript for illustration and validate with the PXRD pattern.

Response 3: Yes, the PXRD testing can give a more robust result, so in the revised version, one example was given in this manuscript for illustration and validation with the PXRD pattern.

Comment 4: In this study, in my opinion, adding ethanol in crystallization system seems no advantages for controlling the polymorph of L-histidine. Why this solvent was selected in this study?

Response 4: Because this research want to study the effect of initial concentration of feed solution on polymorphic crystallization, and the solubility difference was much high at different ethanol fraction, so the ethanol fraction was selected.

Comment 5: If possible, please also give examples to compare particle size and crystal habit of L-histidine from RD and MT crystallizer.

Response 5: Yes, it was a good suggestion. But in this research, we didn’t measure the particle size and take the crystal images of L-histidine from RD and MT crystallizer.

Comment 6: Keywords are missing.

Response 6: Thank you for your suggestion, the keywords was added according to your suggestion.